# Assessing the Causal Association between Biological Aging Biomarkers and the Development of Cerebral Small Vessel Disease: A Mendelian Randomization Study

**DOI:** 10.3390/biology12050660

**Published:** 2023-04-27

**Authors:** Biying Lin, Yuzhu Mu, Zhongxiang Ding

**Affiliations:** 1Department of Radiology, Affiliated Hangzhou First People’s Hospital, Zhejiang University School of Medicine, 261 Huansha Rd., Hangzhou 310006, China; 2Department of Radiology, The Fourth School of Clinical Medicine, Zhejiang Chinese Medical University, Hangzhou 310006, China

**Keywords:** cerebral small vessel disease, biological aging, leukocyte telomere length, epigenetic clock, Mendelian randomization

## Abstract

**Simple Summary:**

Chronological aging has long been considered an immutable risk factor for cerebral small vessel disease, but recent studies have revealed the potential for biological aging to be modulated and possibly even slowed or reversed, offering a promising avenue for intervention. Emerging studies have reported that biological aging biomarkers, such as leukocyte telomere length and epigenetic clocks, are associated with the risk of cerebral small vessel disease. However, the results remain inconsistent and controversial. Hence, in this study, we sought to explicate the underlying causal relationship between biomarkers of aging and cerebral small vessel disease. The results show that genetically determined leukocyte telomere length and epigenetic clocks were not associated with the ten measures of cerebral small vessel disease. Our study did not provide evidence to support a causal association between cerebral small vessel disease and the aging biomarkers of leukocyte telomere length and epigenetic clocks; thus, more studies are needed to evaluate the potential of reverse-biological-aging therapy in cerebral small vessel disease.

**Abstract:**

Biological aging biomarkers, such as leukocyte telomere length (LTL) and epigenetic clocks, have been associated with the risk of cerebral small vessel disease (CSVD) in several observational studies. However, it is unclear whether LTL or epigenetic clocks play causal roles as prognostic biomarkers in the development of CSVD. We performed a Mendelian randomization (MR) study of LTL and four epigenetic clocks on ten subclinical and clinical CSVD measures. We obtained genome-wide association (GWAS) data for LTL from the UK Biobank (N = 472,174). Data on epigenetic clocks were derived from a meta-analysis (N = 34,710), and CSVD data (N cases =1293–18,381; N controls = 25,806–105,974) were extracted from the Cerebrovascular Disease Knowledge Portal. We found that genetically determined LTL and epigenetic clocks were not individually associated with ten measures of CSVD (IVW *p* > 0.05), and this result was consistent across sensitivity analyses. Our findings imply that LTL and epigenetic clocks may not help in predicting CSVD development as causal prognostic biomarkers. Further studies are needed to illustrate the potential of reverse biological aging in serving as an effective form of preventive therapy for CSVD.

## 1. Introduction

Cerebral small vessel disease (CSVD) represents a range of pathological changes affecting the microvascular circulation of the brain, causing 25% of strokes and 45% of dementia cases. Neuroimaging abnormalities in CSVD include white matter hyperintensities (WMH), small subcortical infarcts, lacunes, and microbleeds [1,2,3]. The prevalence of CSVD increases with age, placing a huge burden on societies with an aging population [4]. There are still no prognostic biomarkers or effective preventive therapy for CSVD. Emerging studies have reported that biological aging biomarkers, such as leukocyte telomere length (LTL) and epigenetic clocks, are associated with CSVD risk.

Telomeres are terminal nucleoprotein complexes located at the ends of linear chromosomes, the function of which is to safeguard genomic stability and integrity, and their length gradually decreases over time and plays a direct role in cell senescence [5,6]. Thus, telomere length is widely considered a reliable indicator of cellular aging. Some observational studies suggested that shorter LTL is associated with heavier WMH loading and a higher risk of ischemic or hemorrhagic stroke [7,8,9]. In contrast, other cohort studies indicated that shorter LTL is significantly associated with lower WMH loading and lower rates of lacunar infarction [10]. Furthermore, the results of a meta-analysis recently conducted indicate that there is no significant relationship between LTL and WMH [11].

Epigenetic clocks are derived from mathematical algorithms based on the DNA methylation (DNAm) state of different sets of cytosine–phosphate–guanine sites (CpGs) in the genome [12]. Whether an individual’s epigenetic age is biologically younger or older than their chronological age can be measured using epigenetic age acceleration (EAA) [13]. HannumAge and Intrinsic HorvathAge predict chronological age with remarkable accuracy, while PhenoAge and GrimAge outperform these models for predicting health status and mortality [14,15]. Several studies indicate that GrimAge, HannumAge, and HorvathAge acceleration have a positive correlation with the volume of WMH [16,17]. Conversely, other studies suggest that a negative correlation exists between Horvath epigenetic age acceleration and fractional anisotropy (FA), which is a marker of the integrity of white matter tracts [18]. In contrast, HannumAge has been found to have a positive association with global FA and a negative association with global mean diffusivity (MD), which is sensitive to diffuse white matter injury [19]. Moreover, a cross-sectional study indicated GrimAge was not significantly associated with FA or MD [17]. 

This lack of consensus may be due to the vulnerability of observational investigations to confounding variables and reverse causality. In order to mitigate the biases of these factors, we introduced the Mendelian randomization (MR) method [20]. In the MR study, single-nucleotide polymorphisms (SNPs) of a genome-wide association study (GWAS) were used as instrumental variables (IVs) to increase the reliability of causal inferences. Reverse causation is avoided, as the genotype cannot be modified after conception. Thus, an MR study can be perceived as a natural randomized controlled trial, where the random allocation of mutations to gametes during meiosis replicates the principles of randomization in an experimental setting [21,22]. 

An increasing number of studies have suggested that age-related disease may share the same underlying cause and can be treated or cured through reverse-biological-aging therapy, for instance, by using the Yamanaka factors (or a subset; OCT4, SOX2, and KLF4; OSK) [23,24]. Biological aging biomarkers have the potential to serve as monitoring biomarkers and treatment targets to prevent and treat age-related diseases; however, the phenomenon has yet to be evaluated for CSVD. In this two-sample MR analysis, we aimed to demonstrate whether telomere length and epigenetic clocks have a causal role in the development of cerebral small vessel disease.

## 2. Materials and Methods

### 2.1. Study Design Overview

We used a two-sample MR study to estimate the genetically predicted effects of telomere length and epigenetic clocks (GrimAge, PhenoAge, HannumAge, and Intrinsic HorvathAge acceleration) on ten measures of CSVD (“WMH volume”, “FA”, “MD”, “lacunar stroke”, “all-location brain microbleeds (BMBs)”, “lobar BMBs”, “mixed or deep BMBs”, “all-location intracerebral hemorrhage (ICH) or small vessel stroke (SVS)”, “lobar ICH or SVS”, and non-lobar ICH or SVS) (Figure 1). 

In our MR analyses, three key assumptions were considered when defining the valid instrumental variables [25]. Firstly, in terms of the relevance assumption, the selected instrumental variables (IVs) were considered highly correlated with exposure. Secondly, in terms of the independence assumption, IVs were not considered associated with confounders (e.g., blood pressure, blood glucose, hypertension, diabetes, hyperhomocysteinemia, drinking, and smoking). Lastly, regarding the exclusion restriction assumption, IVs were assumed to influence the outcomes only through the exposure pathway (Figure 2).

### 2.2. Genetic Instruments for LTL and Epigenetic Clocks

The genetic variants associated with LTL were obtained from the published GWAS data of a large sample comprising approximately 472,174 participants of European ancestry from the UK Biobank [26]. We also used publicly available summary statistics data on epigenetic age acceleration from a recent GWAS meta-analysis, which included 34,710 participants of European ancestry from 28 cohort studies. A detailed description of the methods used in this study can be found in a previous publication by McCartney et al. (2021) [27]. The sample characteristics of the study population are described in Appendix A.

### 2.3. Genetic Association Data Sources for CSVD Phenotypes

We obtained ten GWAS parameters of CSVD (“WMH volume”, “FA”, “MD”, “lacunar stroke”, “all-location BMBs”, “lobar BMBs”, “mixed or deep BMBs”, “all-location ICH or SVS”, “lobar ICH or SVS”, and “non-lobar ICH or SVS”) from the Cerebrovascular Disease Knowledge Portal (www.cerebrovascularportal.org). The GWAS indicators of WMH volume, FA, and MD were evaluated in the data from 18,381, 17,663, and 17,467 European participants. WMH volume was log-transformed and normalized for brain volume. FA and MD were reduced to their first-principle components [28]. A meta-analysis was conducted to investigate 6030 cases of lacunar stroke, with 248,929 European participants used as controls; lacunar stroke samples were divided into the magnetic-resonance-imaging (MRI)-confirmed group and TOAST group. In the MRI-confirmed group, lacunar stroke was defined with anatomically compatible lesions on MRIs (subcortical infarct, ≤15 mm in diameter), either as high-intensity regions via diffusion-weighted imaging for acute infarcts or as low-intensity regions through fluid-attenuated inversion recovery or T1 imaging for non-acute infarcts, as well as the absence of causes of stroke other than small vessel disease. In the TOAST group, lacunar stroke was defined according to the TOAST criteria, based on clinical lacunar syndrome and the absence of other causes of stroke or based on the diagnosis of non-lacunar infarction using computed tomography (CT) [29]. The meta-analysis of “all-location BMBs”, “lobar BMBs”, and “mixed or deep BMBs” was conducted in 3556, 2179, and 1293 cases, respectively, using GWAS data from 25,862 participants of European ancestry from 11 cohort studies. BMBs were recognized as small, hypointense lesions on susceptibility-weighted imaging (SWI) sequences or, to a lesser extent, on T2*-weighted gradient echo sequences, and were further differentiated by location [30]. The GWAS meta-analysis of “ICH or SVS” included samples from 241,024 participants, consisting of 6255 ICH or SVS cases and 233,058 control subjects. These cross-phenotype analyses estimated the genetic overlap between the SVS GWAS and ICH GWAS data [31]. The summarized results are listed in Appendix A. More detailed information about sample demographics and methods of analysis are available in the original studies. All the relevant studies for data sources received ethical approval and obtained informed consent from all participants.

### 2.4. MR Analyses

We performed this two-sample MR study based on different GWAS summary results, which could increase its estimated power. The threshold of genome-wide significance was set at *p* < 5 × 10^−8^ for genetic instruments associated with LTL and epigenetic clocks. Although the number of IVs could be increased, this could lead to a lower threshold, and therefore, the additional IVs may be weak instruments, thus decreasing the statistical power. In the meantime, we used the linkage disequilibrium (LD) test to obtain independent SNPs associated with LTL and epigenetic clocks. In addition, in order to avoid the possible pleiotropic effects, we investigated whether SNPs were associated with confounding factors (blood pressure, blood glucose, hypertension, diabetes, hyperhomocysteinemia, drinking, and smoking). We searched all the selected SNPs in the PhenoScanner database (http://www.phenoscanner.medschl.cam.ac.uk/ (accessed on 1 January 2023)) (*p* < 5 × 10^−8^). Then, we used these SNPs to harmonize the ten measures of CSVD data. Alleles and palindromic SNPs were also removed using the same criteria.

In our MR analyses, the IVW method was used as the primary method. A fixed- or random-effect meta-analysis was used together in the IVW method to estimate the causal relationship between exposure and outcome. We chose the random-effect model if there was heterogeneity between SNPs. In the IVW method, the weighted regression of SNP-specific Wald ratios was used to estimate the causal effects while assuming all the IVs were valid. However, if there is horizontal pleiotropy, the estimate can be biased [32]. Thus, the MR-Egger method was used for sensitivity analysis. Even in the case of invalid SNPs, the MR-Egger method could infer a causal relationship between exposure and outcome. In the causal relationship estimation, unlike IVW methods, the MR-Egger regression does not constrain the slope to pass through zero and can be used to detect and correct for bias due to horizontal pleiotropy [33]. Moreover, Cochran’s Q test was used to quantify the heterogeneity when we used the IVW and MR-Egger methods to analyze the causal relationship. The single-SNP analysis and leave-one-out analysis ensured the reliability of the results. Through these analyses, we studied whether a single SNP was the driving factor in the primary causal association by the removal of every single SNP in turn [34]. Furthermore, an MR-PRESSO global test was used to detect horizontal pleiotropy, and an MR-PRESSO outlier test was used to detect the outlier SNPs and investigate the significant differences before and after removing the outlier SNPs [35]. Finally, based on the sample size of the exposure dataset, the number of IVs and F-statistics were calculated to assess the strength of the IVs [36]. All statistical analyses were conducted using R version 4.1.1. The “TwoSampleMR” version 0.5.6 R package was used to perform the Mendelian randomization analysis [37].

## 3. Results

### 3.1. The Causal Effect of LTL on CSVD

In the IVW method, no causal association was observed between LTL and the ten measures of CSVD, as shown in Figure 3; the full results of the other four MR methods are shown in Appendix A. In the sensitivity analysis, neither the MR-PRESSO global test nor the MR-Egger test indicated pleiotropy across SNPs in the causal estimates for the LTL on CSVD (*p* > 0.05) (Appendix A). The leave-one-out and single-SNP analyses revealed no SNPs with strong influences on the causal estimates. As shown in Appendix A, when heterogeneity was detected using Cochran’s Q test (*p* < 0.05), we used the inverse-variance-weighted (IVW) method in a random-effect model instead. The F-statistics for the IVs of LTL were above 10, indicating that these IVs were strong instruments, thus reducing bias in the IV estimates.

### 3.2. The Causal Effect of Epigenetic Clocks on CSVD

Genetically elevated GrimAge, PhenoAge, HannumAge, and Intrinsic HorvathAge acceleration were not causally associated with the ten measures of CSVD in the IVW model (Figure 4). The full results of the causal associations between all epigenetic clocks and CSVD are shown in Appendix A. The causal estimates were confirmed using the sensitivity analysis (Appendix A). The results of the inverse-variance-weighted (IVW) method in the random-effect model were chosen when heterogeneity was detected using Cochran’s Q test (*p* < 0.05), and pleiotropy was assessed using the MR-PRESSO global test or the MR-Egger test. The calculated F-statistics were all above 10 in the causal associations between the genetically predicted epigenetic clocks and CSVD.

## 4. Discussion

In this two-sample MR analysis, we found no evidence in favor of a causal association of telomere length and epigenetic clocks with the development of CSVD based on the large-scale GWAS data.

To the best of our knowledge, this MR investigation is the first study to identify a causal correlation between biological aging biomarkers, such as telomere length and epigenetic clocks, and CSVD. Our results are consistent with a recent meta-analysis indicating no significant association between LTL and brain WMH [11]. Additionally, a cross-sectional study indicated that GrimAge was not significantly associated with FA or MD [17], but most of these studies only focus on the WMH or DTI manifestations of CSVD and do not include other epigenetic clocks. Our findings are in contrast to some observational studies that support a significant association between telomere length, epigenetic clocks, and WMH [7,8,10,16,17]. Observational studies are prone to confounding variables or reverse causation, whereas an MR study can eliminate such biases. In our MR study, the effects of confounding variables were minimized by removing the SNPs related to possible confounders. A recent study provides evidence of possible reverse causality, where the results may have been misleading by suggesting that CSVD may influence the epigenome and not vice versa [19]. Based on our results, the representative biological aging biomarkers of LTL and epigenetic clocks may not have a causal effect on the development of CSVD.

At present, no definitive consensus exists regarding the precise contribution of aging biomarkers, particularly epigenetic clocks, in elucidating the underlying mechanisms of aging and age-related morbidities. Nevertheless, many epidemiological studies have demonstrated that accelerated epigenetic aging is positively associated with a range of age-related disorders, as well as various indicators of age-related conditions [12,38,39]. Further research is necessary to elucidate whether epigenetic clocks have a causal relationship or a merely on-causal correlation with each age-related disorder and to determine whether age-related risk factors are solely or partially mediated through accelerated epigenetic aging in the development of age-related diseases [40]. The unraveling of such pathways could help distinguish between targets for intervention and biomarkers that are informative for diagnosis or prognosis.

This study involves several limitations. Firstly, we only used LTL, GrimAge, PhenoAge, HannumAge, and Intrinsic HorvathAge acceleration to explore the causal relationship between biological aging biomarkers and CSVD, and more evaluation indicators—such as mDNA cortical age or other aging markers, such as lipids, protein glycosylation, and physiological measures—can be used to further evaluate the causal relationship between biological aging and CSVD risk. Secondly, DTI metrics, such as FA and MD, were included in our study as normalized continuous variables; thus, the clinical values of odds ratios (ORs) were limited. Thirdly, once larger GWAS datasets with more genetic instruments are available, it would also be useful to replicate our analyses with a more rigorous assessment of horizontal pleiotropy to confirm our results. Fourthly, this study was conducted using genetic data of those only from European ancestry, so the results may not be applicable to other ethnicities. 

## 5. Conclusions

In conclusion, using information from large genetic consortia, we did not find any causal association between the aging biomarkers of LTL and epigenetic clocks and the development of CSVD. Further investigations into the effect of biological aging biomarkers and the potential of reverse-biological-aging therapy in CSVD are needed.

## Figures and Tables

**Figure 1 biology-12-00660-f001:**
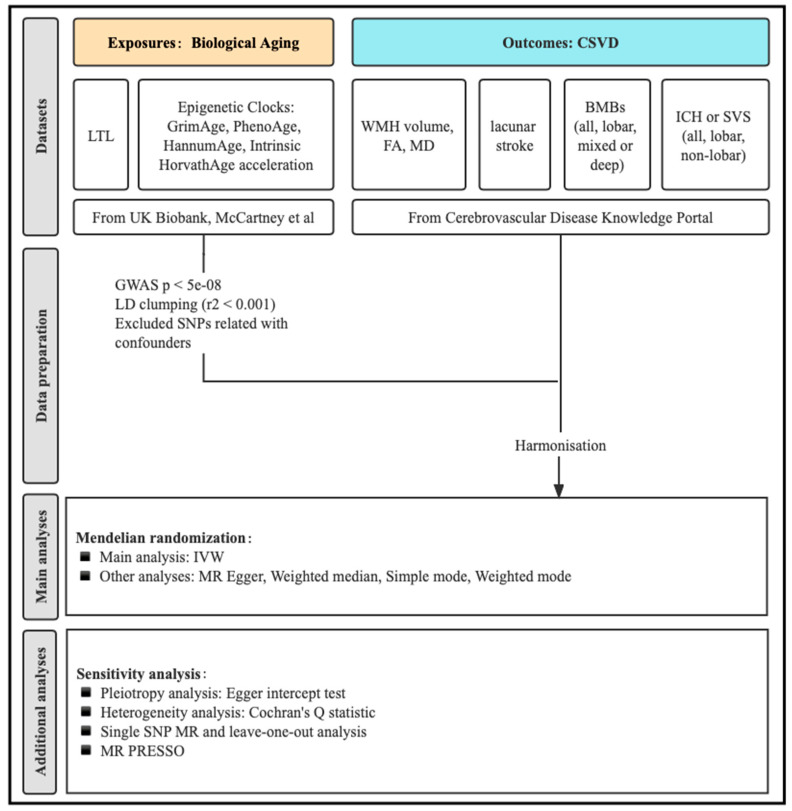
Flowchart of the current study. Exposures are in the yellow box while outcomes are in the blue box.

**Figure 2 biology-12-00660-f002:**
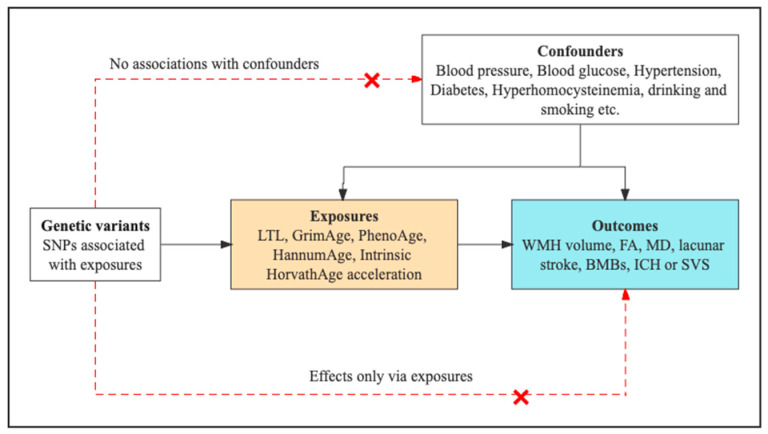
The key assumptions of MR study. The “❌” indicates that genetic variants were not associated with confounders or could not be directly involved in outcomes but rather only via the exposure pathway. Exposures are in the yellow box while outcomes are in the blue box.

**Figure 3 biology-12-00660-f003:**
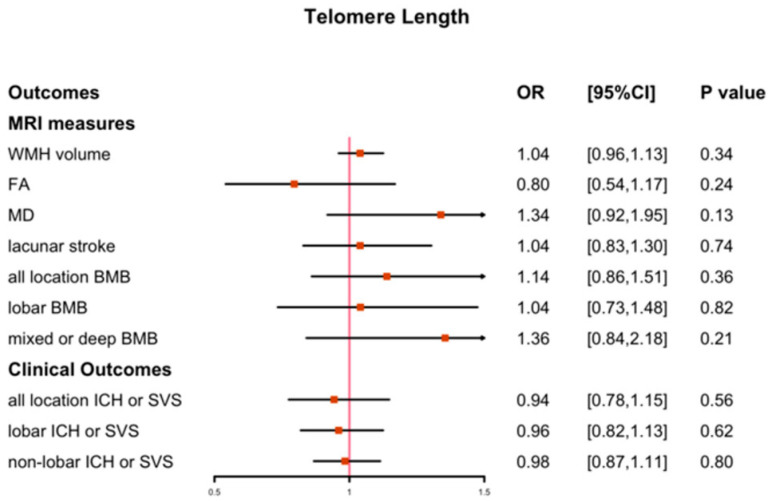
Analysis of MR IVW estimates for genetically predicted effects of LTL on CVSD.

**Figure 4 biology-12-00660-f004:**
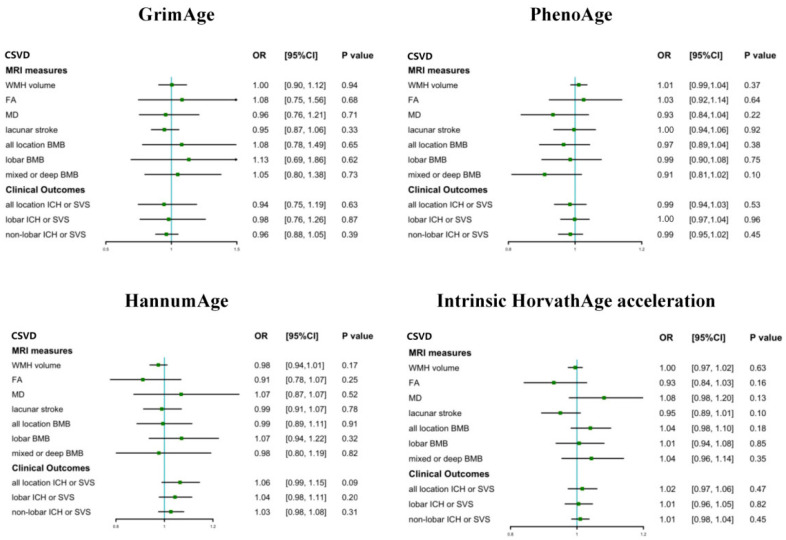
Analyses of MR IVW estimates for genetically predicted effects of epigenetic clocks on CVSD. The green squares in this forest plot show the value of OR while the arrows means the upper 95% CI are upper than the upper limit of the axis.

## Data Availability

Publicly available datasets were analyzed in this study. CSVD GWAS summary data were downloaded from the Cerebrovascular Disease Knowledge Portal (www.cerebrovascularportal.org (accessed on 26 August 2022)). GWAS summary data of LTL were provided by Codd et al. (https://figshare.com/s/caa99dc0f76d62990195 (accessed on 26 August 2022) and McCartney et al. (https://datashare.ed.ac.uk/handle/10283/3645 (accessed on 26 August 2022)).

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
