# Peer review of "Assessing the Causal Association between Biological Aging Biomarkers and the Development of Cerebral Small Vessel Disease: A Mendelian Randomization Study"

_biology, 2023, doi:10.3390/biology12050660_

Round 1

Reviewer 1 Report

This study by Lin et al. utilizes Mendelian Randomization (MR) to investigate whether there is a causal relationship between biomarkers of biological aging such as leukocyte telomere length and epigenetic clocks, and development of cerebral small vessel diseases (CSVD). The novelty as claimed by the authors is that this the first study to utilize MR to comprehensively investigate this causal relationship. The authors use 10 CSVD measures to analyze these relationships. The authors source data from a large number of subjects from various databases but the limitation is that the subjects are all of European descent. The critical finding of this study is that there are no strong causal links between the biomarkers and CSVD measures. Overall, this study is useful in that it establishes that there is no relationship between currently used biomarkers of biological aging and development of CSVD and sheds light on the need for more research to find better biomarkers which might be more predictive of CSVD.

My comments/suggestions are as follows:

1. The authors mention the concept of reversal of biological aging in the simple summary in the beginning as well as in the conclusion at the end, but it is unclear specifically what they are referring to. It would be helpful to discuss the relevance of that to the current study a bit more and to add more information if possible.

2. I have quite a few suggestions for typographical and grammatical edits listed below:

Page 1, Line 12 – “….vessel disease, but recent studies….” instead of “…. vessel disease, recent studies…”

Page 1, Line 13 – “….studies have reported that…” instead of “….studies have been reported that…”

Page 1, Line 20 – “….studies are needed to….” instead of “….studies needed to….”

Page 1, Line 24 – “epigenetic” not “epidenetic”

Page 1, Line 24 – “….play causal roles as…” or “….play a causal role as….” instead of “….play causal role as….”

Page 1, Line 31 and 32 – “….may not be helpful….” instead of “….may not helpful….”

Page 1, Line 32 – “More studies are needed to illustrate….” or “Further studies are needed to illustrate….’ instead of “It needs more studies to illustrate….”

Page 1, Line 35 – Spelling used here and in some other instances is “ageing” whereas “aging” is used in most of the manuscript. Please use one consistent spelling throughout the manuscript.

Page 2, Line 65 – (WMH) not (WHM). This error is found in many places in the manuscript. Please correct this.

Page 2, line 66 – “lacunes” not “launes”

Page 2, Line 66 – “The prevalence of CSVD increases with age, placing a huge burden on any society with an aging population” instead of “CSVD which prevalence increases with age is causing a huge burden on society with the aging population.”

Page 2, Line 68 – “….biomarkers or effective preventive therapy for CSVD.” Instead of “….biomarkers and effective preventive therapy in CSVD.”

Page 2, Line 69 – “….have reported that biological….” Instead of “….have been reported biological….”

Page 2, Line 71 – “chromosomes” instead of “chromosome”

Page 2, Line 72 – “plays” instead of “play”

Page 2, Line 74 – “….associated with heavier….” Instead of “….associated to heavier….”

Page 2, Line 78 – “….relationship between LTL and WMH….” Instead of “….relationship of LTL with WHM…”

Page 2, Line 80 – “….different sets of….” Instead of “….different set of….”

Page 2, Line 80 and 81 – “….in the genome….” Instead of “….in genome….”

Page 2, Line 83 – “….HorvathAge predict chronological….” Instead of “….HorvathAge show predict chronological….”

Page 2, Line 85 – “…. HannumAge, and HorvathAge….” Instead of “….HannumAge, HorvathAge….”

Page 2, Line 86 – “….volume of WMH, while others indicate evidence….” Instead of “….volume of WHM, others indicate that evidence….”

Page 2, Line 88 – “….fractional anisotropy (FA), which is a marker….” Instead of “….fractional anisotropy (FA) that is a marker….”

Page 2, Line 89 – “….shows the….” instead of “….show the the….”

Page 2, Line 90 – “….mean diffusivity (MD), which is sensitive….” Instead of “….mean diffusivity (MD) that is sensitive….”

Page 3, Line 98 – “….causation is avoided as the genotype….” Instead of “….causation could be avoided for the genotype….”

Page 3, Line 99 and 100 – “….since mutations are assigned to gametes randomly when cells undergo meiosis.” Instead of “….when cell undergoes meiosis mutations are assigning to gametes randomly.”

Page 3, Line 109 – “HorvathAge” not “HovathAge”

Page 3, Line 113 – “….assumptions were held when valid instrumental variables had been defined….” Instead of “….assumptions had been held when valid instrumental variables been defined….”

Page 3, Line 114 – “….assumption requires that selected….” Instead of “….assumption require selected….”

Page 3, Line 115 – “….assumption requires that IVs….” Instead of “….assumption require IVs….”

Page 3, Line 118 – “….assumption requires that IVs….” Instead of “….assumption require IVs….”

In Figure 1, HorvathAge is misspelled as HovathAge.

Page 4, Line 141 – “….and TOAST groups. In the….” Instead of “…and TOAST groups, In the….”

Page 4, Line 149 – “….“lobar BMBs”, and “mixed or deep BMBs”….” Instead of “….“lobar BMBs”, “mixed or deep BMBs”….”

Page 4, Line 150 – “….2179, and 1293 cases respectively in….” instead of “….2179, 1293 cases in….”

Page 4, Line 151 – “….studies. BMBs were….” Instead of “….studies, BMBs were….”

Page 4, Line 156 – “….location of SVS GWAS….” Instead of “….location from SVS GWAS….”

Page 5, Line 157 – “detailed” instead of “details”

Page 5, Line 162 – “performed” instead of “perform”

Page 5, Line 169 – “identified” instead of “identify”

Page 5, Line 173 and 174 – “….respectively. Alleles and palindromic SNPs were also removed using the same criteria.” Instead of “….respectively according to the same effect alleles and palindromic SNPs were removed.”

Page 5, Line 175 – “utilized” instead of “conducted”

Page 5, Line 176 – “was” instead of “were”

Page 5, Line 180 – “….valid. However, if there is horizontal pleiotropy….” Instead of “….valid, However, if there are horizontal pleiotropy….”

Page 5, Line 181 – “used for” instead of “performed in”

Page 5, Line 181 and 182 – “….Even if invalid SNPs exist, MR-Egger….” Instead of “….Even invalid SNPs are 181 existed, MR-Egger….”

Page 5, Line 184 – “….zero and can be used to detect and correct for the bias due to horizontal pleiotropy.” Instead of “….zero that is robust to horizontal pleiotropy.”

Page 5, Line 187 – “ensured” instead of “could make sure”

Page 5, Line 187 – “These analyses identified…” instead of “It identified…”

Page 5, Line 188 and 189 – “in turn” instead of “at turn”

Page 5, Line 189 – “was used” instead of “were used”

Page 5, Line 190 – “was used” instead of “were used”

Page 5, Line 199 – “as shown in” instead of “was shown in”

Page 6, Line 217 – “was tested” instead of “were tested”

Page 7, Line 231 – “Besides” instead of “besides”

Page 7, Line 233 – “do not include” instead of “not included”

Page 7, Line 234 and 235 – “…observation studies which support a significant…” instead of “….observation studies, these observational studies support a significant….”

Page 7 Line 236 – “….observational studies are….” Instead of “….the observational study is….”

Page 7, Line 237 – “…and an MR study….” Instead of “….and MR study….”

Page 7, Line 239 – 241 – “A recent study provides evidence of possible reverse causality where the results may have been misleading by suggesting that CSVD may influence the epigenome and not the other way around” instead of “One of recent study may provide the evidence of possible reverse causality that misleading the results of observational studies that CSVD may influence the epigenome and not the other way around.”

Page 7, Line 241 – “Based on….” instead of “Depending on….”

Page 7, Line 248 – Should it be “physiological” instead of “physicological”?

Page 7, Line 254 and 255 – “….results may not be applicable to other ethnicities….” Instead of “….results are frequently unable to apply to other ethnicities….”

Page 7, Line 258 – “epigenetic” instead of “epidenetic”

Author Response

Response to Reviewer 1 Comments

Point 1: The authors mention the concept of reversal of biological aging in the simple summary in the beginning as well as in the conclusion at the end, but it is unclear specifically what they are referring to. It would be helpful to discuss the relevance of that to the current study a bit more and to add more information if possible.

Response 1: Thank you very much for your careful work. I have added the following context in Page 3, Line 109 to Page 4, Line 114: Since more and more studies suggested that age-related disease may share the same underlying cause and can be treated or cured by reverse biological aging therapy like using the Yamanaka factors (or a subset; OCT4, SOX2, and KLF4; OSK) [23,24]. Biological aging biomarkers show the potential being monitoring biomarkers and treatment targets to prevent and treat age-related diseases, though the phenomenon has yet to be evaluated for CSVD.

References

  1. Lu Y.; Brommer B.; Tian X.; et al. Reprogramming to recover youthful epigenetic information and restore vision. Nature. 2020, 588, 124-129.
  2. Yang JH.; Hayano M.; Griffin PT.; et al. Loss of epigenetic information as a cause of mammalian aging. Cell. 2023, 186, 305-326.

Point 2: I have quite a few suggestions for typographical and grammatical edits listed below:

Page 1, Line 12 – “….vessel disease, but recent studies….” instead of “…. vessel disease, recent studies…”

Page 1, Line 13 – “….studies have reported that…” instead of “….studies have been reported that…”

Page 1, Line 20 – “….studies are needed to….” instead of “….studies needed to….”

Page 1, Line 24 – “epigenetic” not “epidenetic”

Page 1, Line 24 – “….play causal roles as…” or “….play a causal role as….” instead of “….play causal role as….”

Page 1, Line 31 and 32 – “….may not be helpful….” instead of “….may not helpful….”

Page 1, Line 32 – “More studies are needed to illustrate….” or “Further studies are needed to illustrate….’ instead of “It needs more studies to illustrate….”

Page 1, Line 35 – Spelling used here and in some other instances is “ageing” whereas “aging” is used in most of the manuscript. Please use one consistent spelling throughout the manuscript.

Page 2, Line 65 – (WMH) not (WHM). This error is found in many places in the manuscript. Please correct this.

Page 2, line 66 – “lacunes” not “launes”

Page 2, Line 66 – “The prevalence of CSVD increases with age, placing a huge burden on any society with an aging population” instead of “CSVD which prevalence increases with age is causing a huge burden on society with the aging population.”

Page 2, Line 68 – “….biomarkers or effective preventive therapy for CSVD.” Instead of “….biomarkers and effective preventive therapy in CSVD.”

Page 2, Line 69 – “….have reported that biological….” Instead of “….have been reported biological….”

Page 2, Line 71 – “chromosomes” instead of “chromosome”

Page 2, Line 72 – “plays” instead of “play”

Page 2, Line 74 – “….associated with heavier….” Instead of “….associated to heavier….”

Page 2, Line 78 – “….relationship between LTL and WMH….” Instead of “….relationship of LTL with WHM…”

Page 2, Line 80 – “….different sets of….” Instead of “….different set of….”

Page 2, Line 80 and 81 – “….in the genome….” Instead of “….in genome….”

Page 2, Line 83 – “….HorvathAge predict chronological….” Instead of “….HorvathAge show predict chronological….”

Page 2, Line 85 – “…. HannumAge, and HorvathAge….” Instead of “….HannumAge, HorvathAge….”

Page 2, Line 86 – “….volume of WMH, while others indicate evidence….” Instead of “….volume of WHM, others indicate that evidence….”

Page 2, Line 88 – “….fractional anisotropy (FA), which is a marker….” Instead of “….fractional anisotropy (FA) that is a marker….”

Page 2, Line 89 – “….shows the….” instead of “….show the the….”

Page 2, Line 90 – “….mean diffusivity (MD), which is sensitive….” Instead of “….mean diffusivity (MD) that is sensitive….”

Page 3, Line 98 – “….causation is avoided as the genotype….” Instead of “….causation could be avoided for the genotype….”

Page 3, Line 99 and 100 – “….since mutations are assigned to gametes randomly when cells undergo meiosis.” Instead of “….when cell undergoes meiosis mutations are assigning to gametes randomly.”

Page 3, Line 109 – “HorvathAge” not “HovathAge”

Page 3, Line 113 – “….assumptions were held when valid instrumental variables had been defined….” Instead of “….assumptions had been held when valid instrumental variables been defined….”

Page 3, Line 114 – “….assumption requires that selected….” Instead of “….assumption require selected….”

Page 3, Line 115 – “….assumption requires that IVs….” Instead of “….assumption require IVs….”

Page 3, Line 118 – “….assumption requires that IVs….” Instead of “….assumption require IVs….”

In Figure 1, HorvathAge is misspelled as HovathAge.

Page 4, Line 141 – “….and TOAST groups. In the….” Instead of “…and TOAST groups, In the….”

Page 4, Line 149 – “….“lobar BMBs”, and “mixed or deep BMBs”….” Instead of “….“lobar BMBs”, “mixed or deep BMBs”….”

Page 4, Line 150 – “….2179, and 1293 cases respectively in….” instead of “….2179, 1293 cases in….”

Page 4, Line 151 – “….studies. BMBs were….” Instead of “….studies, BMBs were….”

Page 4, Line 156 – “….location of SVS GWAS….” Instead of “….location from SVS GWAS….”

Page 5, Line 157 – “detailed” instead of “details”

Page 5, Line 162 – “performed” instead of “perform”

Page 5, Line 169 – “identified” instead of “identify”

Page 5, Line 173 and 174 – “….respectively. Alleles and palindromic SNPs were also removed using the same criteria.” Instead of “….respectively according to the same effect alleles and palindromic SNPs were removed.”

Page 5, Line 175 – “utilized” instead of “conducted”

Page 5, Line 176 – “was” instead of “were”

Page 5, Line 180 – “….valid. However, if there is horizontal pleiotropy….” Instead of “….valid, However, if there are horizontal pleiotropy….”

Page 5, Line 181 – “used for” instead of “performed in”

Page 5, Line 181 and 182 – “….Even if invalid SNPs exist, MR-Egger….” Instead of “….Even invalid SNPs are 181 existed, MR-Egger….”

Page 5, Line 184 – “….zero and can be used to detect and correct for the bias due to horizontal pleiotropy.” Instead of “….zero that is robust to horizontal pleiotropy.”

Page 5, Line 187 – “ensured” instead of “could make sure”

Page 5, Line 187 – “These analyses identified…” instead of “It identified…”

Page 5, Line 188 and 189 – “in turn” instead of “at turn”

Page 5, Line 189 – “was used” instead of “were used”

Page 5, Line 190 – “was used” instead of “were used”

Page 5, Line 199 – “as shown in” instead of “was shown in”

Page 6, Line 217 – “was tested” instead of “were tested”

Page 7, Line 231 – “Besides” instead of “besides”

Page 7, Line 233 – “do not include” instead of “not included”

Page 7, Line 234 and 235 – “…observation studies which support a significant…” instead of “….observation studies, these observational studies support a significant….”

Page 7 Line 236 – “….observational studies are….” Instead of “….the observational study is….”

Page 7, Line 237 – “…and an MR study….” Instead of “….and MR study….”

Page 7, Line 239 – 241 – “A recent study provides evidence of possible reverse causality where the results may have been misleading by suggesting that CSVD may influence the epigenome and not the other way around” instead of “One of recent study may provide the evidence of possible reverse causality that misleading the results of observational studies that CSVD may influence the epigenome and not the other way around.”

Page 7, Line 241 – “Based on….” instead of “Depending on….”

Page 7, Line 248 – Should it be “physiological” instead of “physicological”?

Page 7, Line 254 and 255 – “….results may not be applicable to other ethnicities….” Instead of “….results are frequently unable to apply to other ethnicities….”

Page 7, Line 258 – “epigenetic” instead of “epidenetic”

Response 2: Thank you for your careful work. I have revised all of the above points with yellow highlighted.

Reviewer 2 Report

Reviewer Comments:

The manuscript titled “Assessing the causal role of biological aging biomarkers in the development of cerebral small vessel disease: a Mendelian randomization study" by Biying Lin et al shows a comprehensive conclusion report about the non-involvement association of leukocyte telomere length (LTL) and epigenetic clock with cerebral small vessel disease (CSVD). I appreciate authors for taking up this study and elucidating the importance of biological aging markers in the etiology of CSVD. The manuscript has great merits but encountered minor problems.

1.     Authors need to explain the rationale to choose only these two biological markers for the study while there nearly 40+ biological aging markers available.

2.     Authors nowhere mentioned the Single Nucleotide Polymorphisms (SNPs) in detail particularly played a role in CSVD i.e. there are lot of SNPs played a role in CSVD. Clarity is missing.

3.     English language grammar editing is mandatory throughout the manuscript before accept for publication.

Author Response

Point 1: Authors need to explain the rationale to choose only these two biological markers for the study while there nearly 40+ biological aging markers available.

Response 1: Thank you for your careful work. Our study choosed leukocyte telomere length and epigenetic clocks for the emulating reasons. Firstly, Telomere attrition and epigenetic alterations are grouped in primary category in the scheme of 12 main hallmarkers of aging according to the latest progress in aging field [1]. Secondly, Compared to other hallmarkers of aging, telomere attrition and epigenetic alterations could be quantitatively measured by leukocyte telomere length and four established epigenetic clocks (GrimAge, PhenoAge, HannumAge, HovathAge). Thirdly, Leukocyte telomere length and epigenetic clocks have been found associated with cerebral small vessel disease risk in many observational studies. All up, leukocyte telomere length and epigenetic clocks are suitable biomarkers for biological aging in our study.

References

  1. López-Otín C.; Blasco MA.; Partridge L.; et al. Hallmarks of aging: An expanding universe. Cell. 2023 186, 243-278.

Point 2: Authors nowhere mentioned the Single Nucleotide Polymorphisms (SNPs) in detail particularly played a role in CSVD i.e. there are lot of SNPs played a role in CSVD. Clarity is missing

Response 2: Thank you for your careful work. In our study, Selected SNPs were just used as instrumental variables (IVs) performing the causality derivation between exposures and outcomes. The specific meaning of each SNP has not been focused in current study. Further research may conducted to explore the specific biological signaling pathways of each SNP.

Point 3: English language grammar editing is mandatory throughout the manuscript before accept for publication.

Response 3: Thank you very much for your careful work. We have corrected and improved our English language grammar editing carefully.